# Congenital Isolated Unilateral Agenesis of Pulmonary Arteries with 3-Year Follow-Up after Initiation of Treatment

**DOI:** 10.3390/medicina59061161

**Published:** 2023-06-16

**Authors:** Marohito Nakata, Naoko Yokota, Hiroki Uehara, Kazuiko Tabata, Tsuneaki Kenzaka

**Affiliations:** 1Department of Cardiology, Urasoe General Hospital, Urasoe 901-2132, Japan; xbqhf498@yahoo.co.jp (M.N.); naokoyixx86@yahoo.co.jp (N.Y.); cardioexpham1225@hotmail.com (H.U.); 2Department of Internal Medicine, Naha City Hospital, Naha 902-8511, Japan; 39hiko-naha@nch.naha.okinawa.jp; 3Division of Community Medicine and Career Development, Graduate School of Medicine, Kobe University, Kobe 652-0032, Japan

**Keywords:** unilateral agenesis, pulmonary artery, pulmonary hypertension

## Abstract

*Background and objective*: Unilateral agenesis of pulmonary arteries (UAPA) is a rare disease, with approximately 400 cases reported to date. UAPA is often associated with congenital heart disease, and the uncomplicated form is isolated UAPA, which accounts for approximately 30% of all cases of UAPA. The incidence of pulmonary hypertension due to UAPA has been reported to range from 19 to 44%. There is no consensus treatment for pulmonary hypertension associated with UAPA. We present the first reported case in which a three-drug combination, comprising of iloprost inhalation, riociguat, and ambrisentan, was administered to a patient with UAPA, and was followed-up for 3 years post-diagnosis. *Case presentation*: A 68-year-old Japanese woman presented to our hospital with dyspnea and chest discomfort. She underwent chest radiography, blood tests, and echocardiography; however, the cause of the patient’s symptoms could not be identified. During regular follow-up, an echocardiography 21 months after the initial visit revealed elevated right ventricular pressure (peak tricuspid regurgitation velocity: 5.2 m/s and right ventricular systolic pressure: 120 mmHg) and a diagnosis of pulmonary hypertension was made. Contrast-enhanced computed tomography (CT) of the chest and a pulmonary blood flow scintigram were performed to investigate the cause of pulmonary hypertension, and isolated UAPA was diagnosed. The patient was treated with a three-drug combination of iloprost inhalation, riociguat, and ambrisentan and followed up for 3 years with good therapeutic outcomes. *Conclusions*: We present a case of pulmonary hypertension caused by isolated UAPA. Although rare, this disease can lead to pulmonary hypertension and should be treated cautiously. While there is no consensus regarding the treatment of this disease, a three-drug combination of iloprost inhalation, riociguat, and oral ambrisentan proved effective.

## 1. Introduction

Unilateral agenesis of pulmonary arteries (UAPA) is a rare congenital condition caused by the failure of the sixth aortic arch to connect with the pulmonary trunk during embryologic development; it has a prevalence of 1 in 200,000 people, with approximately 400 cases reported to date [1,2]. Most commonly, UAPA occurs in conjunction with cardiovascular abnormalities, such as congestive heart failure and pulmonary hypertension, which typically manifest in infancy. Conversely, in approximately 13–30% patients it can also occur without any associated cardiovascular anomalies; this is termed isolated UAPA. Patients with isolated UAPA can remain asymptomatic into late adulthood, but usually report symptoms such as dyspnea or chest pain, or suffer from hemoptysis or recurrent infections [1,2]. In cases of pulmonary hypertension, surgical revascularization is recommended during infancy, while medical therapy is recommended for older adult patients who are not deemed suitable for surgery [1,2]. Oxygenation, nitric oxide inhalation, continuous intravenous infusion or inhalation of various prostacyclin derivatives, oral phosphodiesterase type 5 (PDE5) inhibitors, and endothelin receptor antagonists are commonly used. However, there have been few studies regarding the long-term follow-up of the effects of these drugs used alone or in combination. Herein, we present the first reported case in which a three-drug combination of iloprost inhalation, riociguat, and ambrisentan was administered to a patient with UAPA, with a 3-year follow-up period after diagnosis.

## 2. Case Report

### 2.1. Case Presentation

A 68-year-old Japanese woman presented to our hospital complaining of dyspnea and chest discomfort. The patient’s medical history included left putaminal hemorrhage, epilepsy, and bronchial asthma. She had no history of alcohol consumption or smoking. Chest radiography at admission revealed hyperinflation of the left lung and atrophy of the right lung (Figure 1).

PA (posterior–anterior) imaging revealed hyperinflation of the left lung, protrusion of the left second arch, and elevation of the right diaphragm, suggesting right lung atrophy.

Echocardiography indicated normal cardiac function and no tricuspid regurgitation. Blood tests revealed normal levels of N-terminal prohormone of brain natriuretic peptide (NT-proBNP). Despite intensive treatment for bronchial asthma, including an increase in the dose of salmeterol xinafoate and fluticasone propionate inhalants from 50 μg/250 μg to 50 μg/500 μg, the patient’s dyspnea did not improve. Twenty-one months after her initial visit, echocardiography revealed elevated right ventricular pressure with a peak tricuspid regurgitation velocity (TRV) of 5.2 m/s and a right ventricular systolic pressure (RVSP) of 120 mmHg, a tricuspid annular plane systolic excursion (TAPSE) of 14.8 mm, and peak systolic velocity of the tricuspid annulus of 7 cm/s, indicating pulmonary hypertension and right ventricular systolic dysfunction. Subsequently, the patient was admitted to the hospital for detailed investigation of the cause of pulmonary hypertension and appropriate treatment.

### 2.2. Investigation

Upon admission, the patient’s vital signs were within normal range, with a blood pressure of 130/70 mmHg, pulse rate of 88 beats/min, respiratory rate of 20 breaths/min, SpO_2_ of 96% without oxygen supplementation, and body temperature of 36.6 °C. Physical examination revealed increased tone in the pulmonary artery component and edema of lower extremities. Electrocardiogram revealed normal axis and right ventricular hypertrophy with an increased R wave in lead V1 (Figure 2). 

Blood tests revealed elevated NT-proBNP levels (Table 1).

### 2.3. Differential Diagnosis

A pulmonary blood flow scintigram and thoracic contrast CT performed according to the diagnostic procedure for pulmonary hypertension (Guidelines for the Treatment of Pulmonary Hypertension) [3] revealed a complete loss of blood flow in the right lung (Figure 3).

Contrast-enhanced CT revealed that the beginning of the right pulmonary artery was deficient (Figure 4).

Right heart catheterization was conducted; nevertheless, pulmonary artery wedge pressure measurement was omitted due to limited experience with right heart catheterization in cases of unilateral absence of the pulmonary artery and apprehension regarding potential hemodynamic compromise. Table 2 shows the details of the examination results. 

The coronary angiography showed no significant stenosis; however, a collateral blood channel from the conus branch of the right coronary artery to the right lung was observed. Echocardiography revealed no congenital heart disease, and other diseases causing pulmonary hypertension were excluded, leading to the diagnosis of isolated UAPA.

### 2.4. Outcome and Follow-Up

The patient was diagnosed with isolated UAPA with associated pulmonary hypertension. We started home oxygen therapy and an endothelin receptor antagonist (ambrisentan, 5 mg orally once daily) because her 6-min walk was 88 m, and her arterial partial pressure of oxygen (PaO_2_) decreased to 42.3 mmHg during exertion. Despite increasing the dosage of ambrisentan to 10 mg during the hospital stay, the right ventricular pressure did not decrease as measured by echocardiography. The patient was readmitted to the hospital 6 months after starting ambrisentan due to persistent dyspnea and NYHA/WHO functional class III symptoms. 

Results from right heart catheterization are presented in Table 2. The patient was started on continuous intravenous infusion of a prostacyclin derivative (epoprostenol), then switched to inhalation of a prostacyclin derivative (iloprost) and oral PDE5 inhibitor (riociguat). Iloprost was chosen for inhalation due to the patient’s aversion to injectable drugs, and riociguat was selected due to contraindications or precautions with sildenafil and tadalafil, as the patient was concurrently taking carbamazepine for epilepsy. 

After 17 days of triple therapy, RVSP improved to 75 mmHg. Subsequent RVSP values fluctuated, but levels of NT-proBNP remained low (Figure 5).

The patient was found to have pulmonary hypertension 21 months after the first visit and was treated for 35 months. The trends of NT-proBNP levels and estimated RVSP are shown in Figure 5.

## 3. Discussion

### 3.1. First and Second Novelty

UAPA is an infrequent etiology of pulmonary hypertension, and its diagnosis is often delayed due to a lack of specific findings [4]. Recognizing UAPA as a potential cause of pulmonary hypertension is essential to achieve early diagnosis. In this case, we could not diagnose UAPA despite characteristic findings on chest radiography at the time of initial examination. There is no established treatment protocol for pulmonary hypertension caused by UAPA [1,2]. Common medical treatments include oxygenation, nitric oxide inhalation, continuous intravenous infusion or inhalation of prostacyclin derivatives, oral PDE5 inhibitors, and endothelin receptor antagonists; however, few studies have reported the long-term efficacy of these treatments when used as single agents or in combination [1,2]. We followed up this patient with UAPA on a three-drug combination (iloprost inhalation, riociguat, and ambrisentan) for 3 years. Detailed changes in RVSP and NT-proBNP estimated using echocardiography were observed, which is the first report of its kind.

### 3.2. Significance of the First and Second Novelty

UAPA is often associated with congenital heart diseases such as tetralogy of Fallot, atrial septal defect, aortic stenosis, right-sided aortic arch, patent ductus arteriosus, and pulmonary atresia. Isolated UAPA without pulmonary hypertension or congestive heart failure may go undiagnosed and remain asymptomatic [1,2]. Symptoms such as dyspnea on exertion, recurrent lower respiratory tract infections, and hemoptysis can appear in patients with UAPA. The underlying mechanism for dyspnea includes pulmonary hypertension, decreased lung capacity, and increased dead space. 

In contrast, decreased lung volume is caused by secondary pulmonary hypoplasia due to decreased blood flow to the lungs. The increased dead space is caused by bronchiectasis, interstitial fibrosis, and the formation of multiple bras due to cystic changes. The basic pathology is impaired lung parenchymal development and cilia motility, and chronic bronchitis [1,2,4]. At the time of the initial examination, the patient complained of dyspnea and chest discomfort, but there were no laboratory findings suggestive of pulmonary hypertension. Symptoms such as dyspnea and chest pain may appear in the stage of pulmonary vascular disease that does not lead to pulmonary hypertension, so it is necessary to be careful [3]. The age at which UAPA is diagnosed has been reported to be from 1 month to 77 years [5], and the diagnosis may be made at an advanced age. In this case, the patient had a history of bronchial asthma and had undergone chest radiography. However, it took some time before UAPA was diagnosed. Previously reported chest radiography findings of UAPA included elevation of the diaphragm on the affected side, decreased lung volume, and loss of pneumomediastinum [5]. This case had all these characteristic findings; however, we did not recognize UAPA because we did not suspect the possibility of this disease. The incidence of pulmonary hypertension associated with UAPA is 19–44%, and not all patients develop pulmonary hypertension [1,2]. The mechanism of pulmonary hypertension caused by UAPA is thought to involve increased blood flow to the pulmonary arteries. As one pulmonary artery was missing, increased blood flow to the remaining pulmonary artery caused shear stress in the endothelium, which resulted in the release of vasoconstrictors, such as endothelin. Chronic vasoconstriction of the pulmonary arteries leads to remodeling, increased pulmonary vascular resistance, and pulmonary hypertension [1,2,4]. Although UAPA is a rare disease, recognizing it as one of the causes of pulmonary hypertension is important for early diagnosis. There is no established treatment for pulmonary hypertension associated with isolated UAPA. In children, surgical treatment may include anastomosis of the hilar artery of the affected lung to the main pulmonary artery, which increases blood flow and improves lung growth, thereby preventing future pulmonary complications [1,2,4]. However, in adults, especially older adult patients, pulmonary arteries are highly fibrotic [2], and revascularization cannot be performed. Medical treatment includes oxygenation, nitric oxide inhalation, continuous intravenous infusion or inhalation of prostacyclin derivatives, oral PDE5 inhibitors, and endothelin receptor antagonists; however, there are few long-term follow-up data involving the use of these drugs as single agents or in combination. There is a report of a 47-year-old female with UAPA complicated with left heart failure and pulmonary hypertension who was treated with sildenafil and followed up for 1 year [5]. There is another report of a 50-year-old patient with UAPA treated with epoprostenol for 9 months [6]. There is only one reported case of triple therapy (prostaglandin receptor agonist, PDE5 inhibitor, and endothelin receptor antagonist) [7]. We diagnosed pulmonary hypertension associated with UAPA and followed up on TRV and NT-proBNP levels in detail for 3 years after initiating drug therapy to demonstrate the usefulness of drug therapy.

### 3.3. Reference to Clinical Utility

Although UAPA is a rare disease, recognizing it as one of the causes of pulmonary hypertension is important for early detection. The three-drug combination therapy of iloprost inhalation, riociguat, and ambrisentan can effectively treat pulmonary hypertension associated with isolated UAPA.

## 4. Conclusions

We report a case of pulmonary hypertension caused by isolated UAPA. Although it is a rare disease, it can cause pulmonary hypertension and should be treated cautiously. Consensus on the treatment for this disease is lacking; however, a three-drug combination of iloprost inhalation, riociguat, and oral ambrisentan can be effective.

## 5. Teaching Point 

Congenital conditions may manifest in advanced age, as observed in this particular case. With the continuous progress of medical advancements leading to an augmented average life expectancy, a surge in the detection of congenital diseases among older adults is anticipated in the future. Consequently, it is advisable to include congenital diseases in the roster of differential diagnoses.

## Figures and Tables

**Figure 1 medicina-59-01161-f001:**
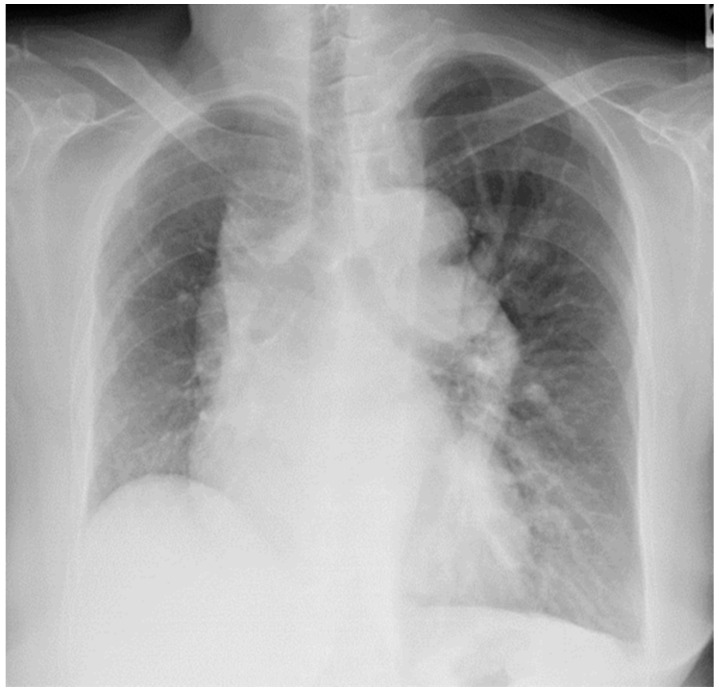
Chest radiograph acquired at the initial visit.

**Figure 2 medicina-59-01161-f002:**
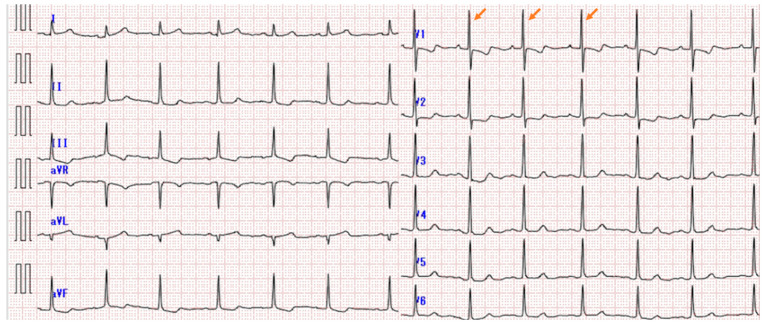
Electrocardiogram at the initial visit. Normal axis and right ventricular hypertrophy, with an increased R wave in lead V1, are indicated by orange arrows.

**Figure 3 medicina-59-01161-f003:**
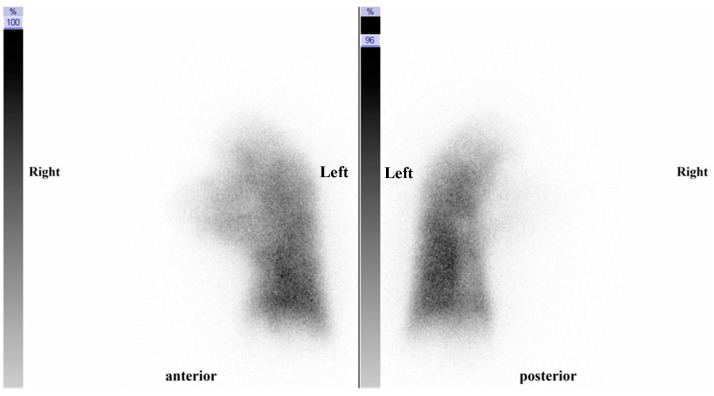
Pulmonary perfusion scintigraphy revealing complete loss of blood flow in the right lung.

**Figure 4 medicina-59-01161-f004:**
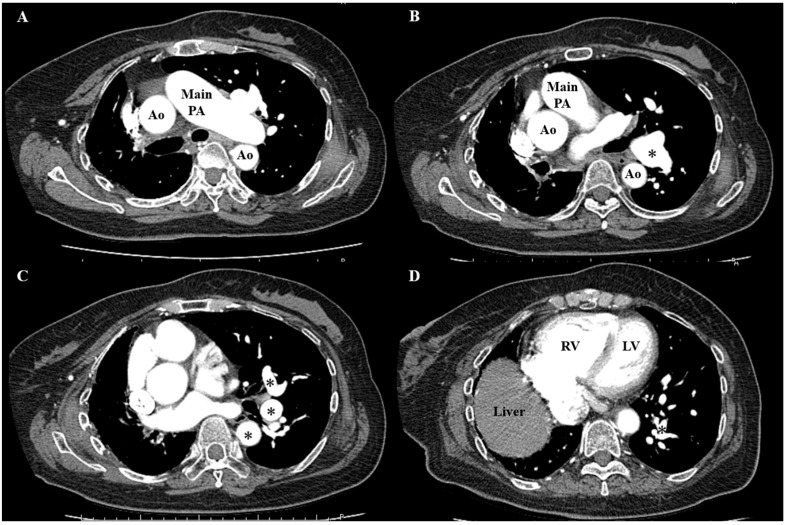
Chest computed tomography with contrast media. (**A**) The right pulmonary artery was deficient from ostium; (**B**) the left pulmonary artery was dilated; (**C**) heart displacement due to decreased right lung volume; and (**D**) the RV was dilated. Ao, aorta; PA, pulmonary artery; RV, right ventricular; LV, left ventricular; *, dilated left pulmonary artery.

**Figure 5 medicina-59-01161-f005:**
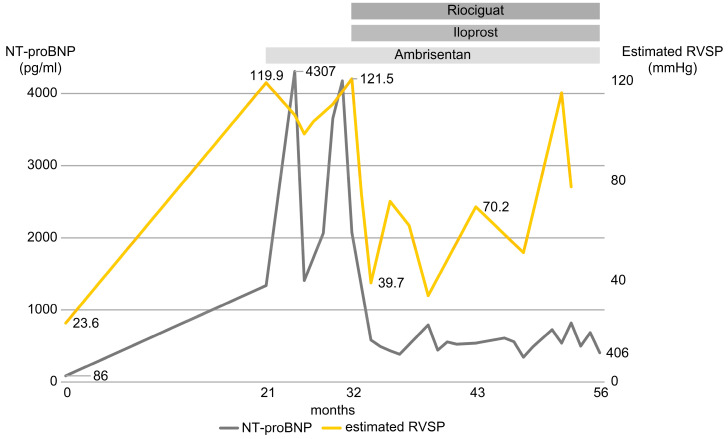
Timeline of therapy with ambrisentan, iloprost, and riociguat, and trends of NT-proBNP and RVSP levels. Yellow line shows RVSP and gray line shows NT-proBNP. NT-proBNP: N-terminal prohormone of brain natriuretic peptide. RVSP: right ventricular systolic pressure.

**Table 1 medicina-59-01161-t001:** Laboratory investigations at initial visit.

Parameter	Value at Initial Visit	Value at First Admission	Normal Range
White blood cell count (cells/µL)	6200	7500	3300–8600
Hemoglobin (g/dL)	12.6	12.5	11.5–15.0
Platelet count (/µL)	19.4 × 10^4^	24.1 × 10^4^	15–35 × 10^4^
C-reactive protein (mg/dL)	1.12	-	≤0.14
Total protein (g/dL)	7.4	6.7	6.6–8.1
Albumin (g/dL)	3.7	3.4	4.1–5.1
Total Bilirubin (mg/dL)	0.6	-	0.4–1.5
Aspartate aminotransferase (U/L)	15	17	13–30
Alanine aminotransferase (U/L)	6	6	7–23
Lactase dehydrogenase (U/L)	180	205	124–222
γ-Glutamyl Transpeptidase (U/L)	53	88	9–32
Blood urea nitrogen (mg/dL)	7.7	11.6	8–20
Creatinine (mg/dL)	0.71	0.79	0.46–0.79
Sodium (mEq/L)	141	138	138–145
Potassium (mEq/L)	4.0	3.9	3.6–4.8
Chloride (mEq/L)	109	104	101–108
Glucose (mg/dL)	93	100	73–109
NT-proBNP (pg/dL)	86	1338	0–125
D-dimer (µg/mL)	-	0.3	0.0–1.0

NT-proBNP: N-terminal prohormone of brain natriuretic peptide.

**Table 2 medicina-59-01161-t002:** Results of right heart catheterization.

Measured Variables	Value at First Admission	Value at Second Admission	Normal Range
Mean right atrial pressure (mRAP) (mmHg)	5	10	2–6
Systolic pulmonary artery pressure (sPAP) (mmHg)	100	108	15–30
Diastolic pulmonary artery pressure (dPAP) (mmHg)	32	37	4–12
Mean pulmonary artery pressure (mPAP) (mmHg)	56	62	8–20
Mean pulmonary artery wedge pressure (mPAWP) (mmHg)	-	14	≦15
Cardiac output (CO) (L/min)	2.94	3.26	4–8
Mixed venous oxygen saturation (SvO_2_)	-	57.5%	65–80%
Systemic blood pressure (mmHg)	126/70	120/57	120/80
Calculated parameters			
Pulmonary vascular resistance (PVR) (WU)	-	14.72	0.3–2.0
Pulmonary vascular resistance index (PVRI) (WU m^2^)	-	20.78 WU m^2^	3–3.5
Total pulmonary resistance (TPR) (WU)	19.0	19.0	<3
Cardiac index (CI) (L/min/m^2^)	2.02	2.31	2.5–4.0
Stroke volume (SV) (mL)	43.3	51.09	60–100
Pulmonary arterial compliance (PAC)	0.64 mL/	0.72 mL/	>2.3 mL/

WU, wood units. The body surface area (BSA) of this patient was 1.41 m^2^. PVR = (mPAP − PAWP)/CO; PVRI = (mPAP − PAWP)/CI; CI = CO/BSA; and PAC = SV/(sPAP − dPAP).

## Data Availability

All data generated or analyzed during this study are included in this published article.

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
