# Peer review of "Congenital Isolated Unilateral Agenesis of Pulmonary Arteries with 3-Year Follow-Up after Initiation of Treatment"

_medicina, 2023, doi:10.3390/medicina59061161_

Round 1

Reviewer 1 Report

73 - this is pseudo-precision. Consider reporting 120 mmHg.

84 - Repeated from line 73

Figure 3

 not all readers may be familiar with reading lung scintigraphy - panels should be marked A, B and mentioned - A - anterior view, B - posterior view

Additionally add L and R on each frame to mark Left lung and Right lung -

Figure 4

This beatiful CT volume rendering does not show the issue. Please crop it so it does.

108 Please confirm the statement „pulmonary artery wedge pressure measurement was not performed due to concerns about hemodynamic compromise.”

I am not sure that wedging one small pulmonary branch would lead to a hemodynamic compromise / collapse

Table 2

156.48 WU 

1662 WU m2 

are you sure about this values - they seem exorbitant

what is the patients hight weight BSA? (state what formula was used)

Cardiac index is calculated by dividing CO by BSA - thus in most adults CI is less than CO. The unit is L/min/m2

181 - 77 is repeated; the second part of the sentence is unnecessary

Side note - 

Mean right atrial pressure (mRAP) of 10 is possibly higher than LA pressure. AFR implantation can be considered as next step interventional paliation

Author Response

Reviewer 1

Response: Thank you for taking the time and effort to review our paper.

We appreciate your positive comments.

  • Line 73: this is pseudo-precision. Consider reporting 120 mmHg.

Response: Thank you for this suggestion. We have corrected the value to 120 mmHg.

  • Line 84: Repeated from line 73.

Response: Thank you for pointing this out. We have removed the following repeated text from the manuscript:

 “Echocardiography revealed a TRV of 5.2 m/sec, RVSP of 119.9 mmHg, tricuspid annular plane systolic excursion (TAPSE) of 14.8 mm, and peak systolic velocity of the tricuspid annulus of 7cm/sec, indicating pulmonary hypertension and right ventricular systolic dysfunction.

  • Figure 3: not all readers may be familiar with reading lung scintigraphy - panels should be marked A, B and mentioned - A - anterior view, B - posterior view

Additionally add L and R on each frame to mark Left lung and Right lung.

Response: Thank you very much for this advice. We have made the corrections as you have suggested in the Figure 3 legend.

Lines 106-108: “Figure 3. Pulmonary perfusion scintigraphy revealing complete loss of blood flow in the right lung.”

  • Figure 4: This beatiful CT volume rendering does not show the issue. Please crop it so it does.

Response: Thank you for the comment. The CT image has been replaced with a revised version that clearly exhibits the anomaly in the right pulmonary artery, accompanied by additional annotations. (Lines 109-116)

Figure 4. Chest computed tomography with contrast media.

(A) The right pulmonary artery was deficient from ostium; (B) The left pulmonary artery was dilated; (C) Heart displacement due to decreased right lung volume; (D) RV was dilated.

Ao, Aorta; PA, Pulmonary artery; RV, Right ventricular; LV, Left ventricular.

*, dilated left pulmonary artery.

  • Lines 108: lease confirm the statement “pulmonary artery wedge pressure measurement was not performed due to concerns about hemodynamic compromise.” I am not sure that wedging one small pulmonary branch would lead to a hemodynamic compromise / collapse

Response: Thank you for this query. I harbored significant concerns due to my lack of prior experience in performing right heart catheterizations for unilateral absence of pulmonary artery (UAPA). I apprehensively recognized the potential for hemodynamic compromise. Consequently, I have diligently revised the text accordingly.

  • Table 2: 156.48 WU 1662 WU m2are you sure about this values - they seem exorbitant

    what is the patients hight weight BSA? (state what formula was used)

    Cardiac index is calculated by dividing CO by BSA - thus in most adults CI is less than CO. The unit is L/min/m2

Response: This section holds paramount significance. I sincerely appreciate your astute observation, as it has brought to light an error in my calculation. We have rectified not only the erroneous value but also included the accompanying formula for precise reference.

  • Line 181: 77 is repeated; the second part of the sentence is unnecessary.

Response: Thank you for pointing this out. We have deleted repeated text.

Reviewer 2 Report

Dear authors,

It has been a pleasure to review this manuscript. Extremely well-written, concise and easy to follow. The Materials and Methods part is excellent.

I have only some minor points to be reviewed:-

The Connclusion part should be 4. not 5.

Maybe you can add some arrows on the ECG to highlight the modifications (increased R wave in V1)

You might add an Abbreviation section after the conclusion section

It might be interesting to add a section: Teaching point..

Author Response

  • The Conclusion part should be 4, not 5.

Response: Thank you for your positive comments and suggestions, and for pointing out this error. We have corrected the section number to 4.

  • Maybe you can add some arrows on the ECG to highlight the modifications (increased R wave in V1).

 Response: Thank you for this suggestion. We have added arrows to the ECG image.

  • You might add an Abbreviation section after the conclusion section.

Response: Thank you for this suggestion. We have added an Abbreviation section. (Lines 241-245)

Abbreviation

UAPA, Unilateral agenesis of pulmonary arteries. CT, computed tomography. PDE 5, phosphodiesterase type 5. NT-proBNP, N-terminal prohormone of brain natriuretic peptide. TRV, tricuspid regurgitation velocity. RVSP, right ventricular systolic pressure. TAPSE, tricuspid annular plane systolic excursion.

  • It might be interesting to add a section: Teaching point.

Response: Thank you for this suggestion. We have added a Teaching point section. (Lines 234-239)

  1. Teaching point

Congenital conditions may manifest in advanced age, as observed in this particular case. With the continuous progress of medical advancements leading to an augmented average life expectancy, a surge in the detection of congenital diseases among the older adults is anticipated in the future. Consequently, it is advisable to include congenital diseases in the roster of differential diagnoses.